# Spatio-Temporal Assessment of Landscape Ecological Risk and Associated Drivers: A Case Study of the Yellow River Basin in Inner Mongolia

Hengrui Zhang [1], Jianing Zhang [1], Zhuozhuo Lv [1], Linjie Yao [1], Ning Zhang [1] and Qing Zhang [1,2,*]

[1] Key Laboratory of Ecology and Resource Utilization of Mongolian Plateau, Ministry of Education, College of Ecology and Environment, Inner Mongolia University, Hohhot 010021, China; 32115166@mail.imu.edu.cn (H.Z.); 32015194@mail.imu.edu.cn (J.Z.); 32215113@mail.imu.edu.cn (Z.L.); 32015108@mail.imu.edu.cn (L.Y.); 22215007@mail.imu.edu.cn (N.Z.)
[2] Collaborative Innovation Center for Grassland Ecological Security, Hohhot 010021, China
* Correspondence: qzhang82@imu.edu.cn; Tel.: +86-13674780584

**Abstract:** The Yellow River Basin in Inner Mongolia (YRBIM) has witnessed major changes in land use/land cover (LULC), which have had an impact on the basin's ecosystem, in the context of fast economic development and urbanization. This study set out to investigate the ecological risk and key driving forces in the basin as LULC evolves. In order to evaluate the ecological risk of the basin and use a geographic detector model to understand the causes of its spatial heterogeneity, we built a landscape ecological risk index (ERI) model based on changes in LULC from 1990 to 2020. The findings indicate that between 1990 and 2020, LULC modifications led to the transfer of several land types to a small number of land types, all of which have since changed into other land types. With high risk areas primarily located in the Hobq Desert, the Hetao irrigation area, and some portions of the Mu Us Sandy Land, the ecological risk level in the basin is gradually decreasing. Human activities are the main cause of the regional variation of ecological risk in the basin, with topography and climate coming in second and third. The Yellow River Basin's ecological danger and environmental quality have only received a limited amount of analysis to date. This study is a crucial resource for the development of civilization and ecological restoration in the region.

**Keywords:** ecosystem risk assessment; land use and land cover changes; landscape pattern; geographical detector; Yellow River Basin in Inner Mongolia

## 1. Introduction

Human social productivity and material levels have rapidly increased in recent years, which has led to more serious worldwide ecological and environmental concerns [1]. Through actions including the reclaiming of farmland, deforestation, overgrazing, the usage of chemical fertilizers and herbicides, urbanization, and air pollution emissions, human activities have changed the natural ecological environment [2,3]. A number of ecological disasters, which include desertification, degradation of land quality, loss of biodiversity, and global climate change, have been caused by these actions [3,4]. These ecological crises have exacerbated ecological risks and have become important factors, affecting national security, restricting economic sustainability, and healthy social development [5,6]. Ecological risk refers to the risks it bears as a result of human activity and the natural environment, as well as its ability to maintain its basic structure and function when disturbed by external factors [7]. As ecological risk assessment evolved in developed countries such as the USA and Europe in the 1970s, it focused on the relationship between human health and environmental pollution, and in 1990, some American scholars began to combine ecological risk assessment with regional landscape studies [7]. Since 2000, ecological risk assessment has been integrated with landscape ecology, and the evaluation objects and evaluation

factors have become more complex [7]. Ecology's heterogeneity assessment is an important research topic because it can uncover the driving factors and processes that maintain ecosystem stability [6]. The field has also promoted qualitative and quantitative research on natural and human factors.

Ecological risk assessment in earlier years concentrated on a single source of risk or a single receptor and used quantitative models to assess ecological risk. Physical models based on entropy and exposure-response methods [8], probabilistic statistical analysis and mechanistic models based on mathematical models [9], and computer models of artificial neural systems are examples [10]. At present, there are currently two predominant approaches for assessing ecological risk: one involves a risk assessment model based on ecological risk sources, using the inherent model of "risk source identification-receptor analysis-exposure and risk characterization" to construct an ecological risk evaluation index system in terms of risk source intensity, receptor exposure, and risk effects [11]. The other method is to build a landscape ecological risk assessment model based on the landscape ecological process and its spatial pattern distribution changes and use remote sensing and GIS technology to analyze the ecological risk situation [12,13]. Landscape ecological risk is defined as the potential adverse consequences of the interaction between landscape patterns and ecological processes under the influence of natural or human-induced factors and can directly reflect the adverse impacts of land use/land cover (LULC) changes on landscape components, structure, and function [14]. There have been many studies worldwide on the relationship between landscape pattern indices, ecological risk, and evaluation methods [15,16]. Among these, a landscape ecological risk assessment model based on LULC data and a landscape index is widely used [17]. A landscape ecological risk assessment model with a landscape ecology angle is more appropriate for assessing the environmental risks caused by human interventions than risk assessment models based on ecological risk sources [15]. Landscape heterogeneity, resilience, stability to disturbances, and ecosystem diversity are closely related [18], all of which emphasize the importance of landscape as an element of ecological risk assessment. Landscape ecological risk assessment focuses more on the spatial and temporal heterogeneity of ecological risks than traditional ecological risk assessment methods [19]. Therefore, in this paper, a landscape model-based ecological risk assessment has been chosen to evaluate risks within the basin.

Factors driving landscape ecological risks are currently a research hotspot. These factors include human activities, climate, and topography that interfere with changes in ecological risk in basins [20]. Some scholars have found that the rapid growth of the human footprint has accelerated changes in LULC types, which has assumed an important role in the process of inter-transfer between land types. This has put the basin under enormous pressure and has seriously threatened the ecological recovery of the basin [21]. Climate determines the distribution of heat and moisture at the basin scale, which in turn affects landscape pattern changes and the spatial heterogeneity of ecological risks [22]. High-altitude and high-slope areas are generally less disturbed by human activities, resulting in lower landscape fragmentation levels [23]. Common methods for studying the driving factors of ecological risks include boosted regression tree [17], c correlation analysis [24], and geographic detector [25]. Geographic detection is a method that can detect both numerical and qualitative data as well as the interaction of two factors on the dependent variable [26]. As opposed to other methods, the geographic detector method can better test influencing factors and interpret their interactions [27].

The effective ecological protection of the Yellow River Basin is of great significance for the social and economic development of China [28]. The natural climate conditions of the YRBIM are relatively poor, and previous regional development and unreasonable governance have led to the degradation of the ecological barrier function of the basin [29]. Therefore, this study analyzed the LULC changes in the Inner Mongolia region of the Yellow River Basin from 1990 to 2020. An ecological risk assessment was performed according to the LULC changes, and the contributions of human activities, climate, and topography to

the changes in ecological risk in the basin were quantified. The objective of this study was to comprehensively assess the ecological environment of the YRBIM and provide financial guidance for the sustainable ecological management of the region.

## 2. Materials and Methods

### 2.1. Study Area

The YRBIM is located in the southwest of the Inner Mongolia Autonomous Region of China (37°37′–41°50′ N, 106°20′–112°47′ E), covering a total area of about 14,718.86 × 10² km². The basin runs through the Inner Mongolian Plateau at an altitude of 848–2350 m (Figure 1). The temperate continental climate that characterizes the region, the average annual precipitation in the region is 305 mm and the average annual temperature is 6.5 °C, is primarily concentrated in July, August, and September. The length of the Yellow River in the basin is 843.5 km, accounting for 18.48% of the total length of the Yellow River. The YRBIM features a complex and diverse landscape, with fertile land dominated by croplands on both banks of the river. Furthermore, the basin includes other landforms, such as mountains, deserts, grasslands, and forests [30].

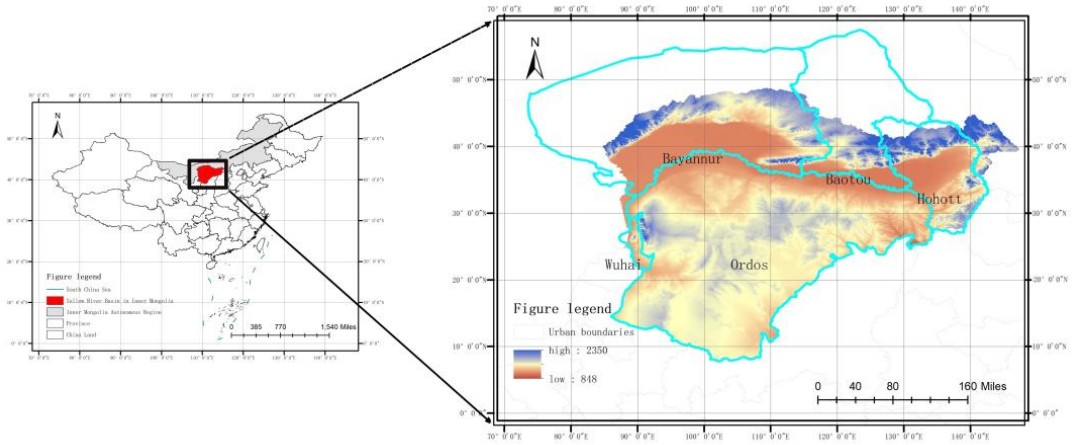

**Figure 1.** The location and topography of the YRBIM.

### 2.2. Analysis of LULC Change in the YRBIM

To explore the LULC changes in the YRBIM from 1990 to 2020, considering the long time series and accessibility of the United States Geological Survey (USGS) remote sensing data, we used Landsat5 TM and Landsat8 OLI images at 30 m resolution from the United States Geological Survey (USGS) in 1990, 1995, 2000, 2005, 2010, 2015, and 2020 (https://www.usgs.gov/ (accessed on 20 May 2023)). LULC data for the YRBIM were produced using images with less than 5% cloud cover from June to September, and data preprocessing, such as cropping and stitching, was conducted using the Google Earth Engine platform. Six wavelength bands (blue, green, red, NIR, and two shortwave IR) were used for feature classification. The normalized vegetation index (NDVI), modified normalized moisture index (MNDWI), and normalized build index (NDBI) were used as band information to improve classification accuracy. The GPS field survey data and sample point data that were selected using Google Earth Pro software were imported into the Google Earth Engine (GEE) platform, of which 70% were training samples and 30% were test samples [31]. Based on the actual situation of the YRBIM, the LULC in the study area was divided into eight types: cropland, grassland, river, lake, swamp, construction land, forest, and bare land. After the overall accuracy assessment and kappa coefficient validation, the overall accuracy of the seven LULC classification results was above 80%, and although there were some potential errors, they could generally meet the needs of the study. Using the LULC data, ArcGIS 10.2 was used to calculate the area of each category in the YRBIM from 1990 to 2020,

construct a land-use transfer matrix, and analyze the LULC changes in the study area over the period from 1990 to 2020 [32].

### 2.3. Analysis of Spatial and Temporal Changes in Ecological Risk in the YRBIM

To analyze the spatial and temporal changes in ecological risk in the YRBIM, we used the landscape ERI as the assessment endpoint. The calculation formula was as follows:

$$ERI = \sum_{i=1}^{n} \frac{A_{ki}}{A_k} R_i \tag{1}$$

where ERI is the ecological risk index for each grid, $A_{ki}$ is the area of the $i$th LULC type in the $k$th grid (km$^2$), $A_k$ is the total area of the $k$th grid (km$^2$), and $R_i$ is the $i$th LULC type of the landscape loss index, calculated as follows:

$$R_i = F_i \times S_i \tag{2}$$

Here, $F_i$ is the ecological vulnerability index, which describes the vulnerability of an ecosystem caused by strong external disturbances resulting from human activities. When the vulnerability is low, the risk to ecosystems is also low. The $F_i$ values of the eight LULC types were as follows: 1 for construction land, 2 for forest, 3 for grassland, 4 for cropland, 5 for lake, 5 for swamp, 5 for river, and 6 for bare land [33], $S_i$ denotes the landscape disturbance index for the $i$th LULC type [34], calculated as follows:

$$S_i = aC_i + bN_i + cD_i \tag{3}$$

Here, $C_i$ stands for landscape fragmentation, $N_i$ for landscape isolation, and $D_i$ for landscape dominance. The weights of $C_i$, $N_i$, and $D_i$ are represented by $a$, $b$, and $c$, respectively, and reflect the impact of anthropogenic disturbance on the ecosystem ($a + b + c = 1$). Depending on the extent of their effects, $a$, $b$, and $c$ were assigned values of 0.5, 0.3, and 0.2, respectively [35].

$C_i$ reflects the change process of landscape structure, function, and ecology and is calculated as follows:

$$C_i = \frac{n_i}{A_i} \tag{4}$$

Here, $Ai$ is the area of the $i$th LULC type (km$^2$) and $n_i$ is the number of patches of the $i$th LULC type.

$N_i$ represents the degree of separation of different patches in the landscape: the higher the number, the more dispersed the landscape and the more the complexity of the landscape distribution. The calculation formula is as follows:

$$N_i = \frac{A}{2A_i} \sqrt{\frac{n_i}{A}} \tag{5}$$

where $A$ is the total study area (km$^2$), and $n_i$ and $A_i$ are given by Equation (4).

$D_i$ reflects the extent to which the patch shape affects the ecological processes within the patch, with larger values indicating a more complex patch shape. It is calculated as follows [36]:

$$D_i = \frac{2\ln(P_i/4)}{\ln A_i} \tag{6}$$

where $Pi$ is the perimeter of the $i$th landscape in the grid.

Using the fishnet tool in ArcGIS 10.2, we divided the study area into 9547 grids with a width and height of 4 km. The 1990–2020 ERI values in each grid [37] were calculated [12] using the kriging interpolation tool in ArcGIS 10.2.

To determine the spatial distribution of different risk levels within the YRBIM, ecological risks were classified into five levels based on the natural breakpoint method using ArcGIS 10.2: lowest risk area (ERI ≤ 0.0195), lower risk area (0.0195 < ERI ≤ 0.0235), middle

risk area ($0.0235 < \text{ERI} \leq 0.0275$), higher risk area ($0.0275 < \text{ERI} \leq 0.0315$), and the highest risk zone ($\text{ERI} > 0.0315$).

To understand whether there is an aggregation effect of ecological risks in the YRBIM, spatial autocorrelation analysis was chosen to reflect the similarity of ERI values of spatially adjacent or nearby units [38]; it was calculated as follows:

$$I = \frac{\sum_{i=1}^{n} \sum_{i=1}^{m} w_{ij}(x_i - \bar{x})(x_j - \bar{x})}{(x_i - \bar{x})^2 \sum_{i=1}^{n} \sum_{i=1}^{m} W_{ij}} \tag{7}$$

where $x_i$ and $x_y$ are the values of the variables in adjacent paired spatial units, $W_{ij}$ is the spatial weight matrix, and $\bar{x}$ is the mean of the attribute values. Moran's $I$ index was calculated using ArcGIS 10.2 to measure the global autocorrelation of ecological risk. When Moran's I index is greater than 0, the results have a significant positive correlation; when Moran's I index is less than 0, the results have a significant negative correlation.

To understand whether there are areas of high-value aggregation (hot spots) and low-value aggregation (cold spots) in the YRBIM for ecological risk indices, cold and hot spot analysis (Getis-Ord Gi*) was chosen to identify the spatial distribution of cold and hot spots [39]. The calculation equation is as follows:

$$G_i* = \frac{\sum_{j=1}^{n} w_{ij} x_j - \bar{x} \sum_{j=1}^{n} w_{ij}}{\left[\sqrt{\frac{\sum_{j=1}^{n} x_j^2}{n-1}}\right] \sqrt{\frac{n \sum_{j=1}^{n} w_{ij}^2 - (\sum_{j=1}^{n} w_{ij})^2}{n-1}}} \tag{8}$$

$$\bar{x} = \frac{1}{n} \sum_{j=1}^{n} x_i \tag{9}$$

where $x_i$ and $x_j$ are the observed values of cells '$i$' and '$j$', $w_{ij}$ is the spatial weight between $x_i$ and $x_j$, $\bar{x}$ is the mean of the landscape ecological risk index, and n is the number of grid cells. $G_i*$ is the local autocorrelation index for the area $i$. The larger the absolute value of $G_i*$, the more statistically significant it is, indicating that the result cannot be generated randomly. When $G_i* > 0$, it means that the area is a hot spot area with high-value aggregation; when $G_i* < 0$, it means that the area is a cold spot area with low-value aggregation; when $G_i* = 0$, it means that the result is not statistically significant when it is randomly generated. The cold hotspot analysis tool of ArcGIS 10.2 was used to analyze the obtained $G_i*$ values, and significance tests were conducted to obtain the cold hotspot areas with confidence intervals and to determine their spatial clustering locations.

*2.4. Analysis of Spatial Heterogeneity Driving Ecological Risk in the YRBIM*

The spatial and temporal variability of ecological risk is mainly driven by a combination of human activities, climate, and topography. In this paper, six drivers, namely, DEM digital elevation model, slope, annual precipitation, population density, NDVI, and anthropogenic disturbance index, were selected to drive the spatial heterogeneity of ecological risk in the YRBIM. The digital elevation model (DEM) with a spatial resolution of 30 m was downloaded from the Geospatial Data Cloud website of the Computer Network Centre of the Chinese Academy of Sciences (https://www.gscloud.cn/ (accessed on 20 May 2023)); slope data were extracted from the DEM data in GIS; annual precipitation data were downloaded from the precipitation data set of Zhai and Zhu's team at Hehai University (http://www.scidb.cn/cstr/31253.11.sciencedb.01607/ (accessed on 20 May 2023)); population density data were downloaded from the Worldpop global population dataset (https://www.worldpop.org/ (accessed on 20 May 2023)); and Normalized Difference Vegetation Index(NDVI) data were downloaded from the Chinese Academy of Sciences Data Centre for Resource and Environmental Sciences (CAS) (http://www.resdc.cn/ (accessed on 20 May 2023)).

To analyze the impact of anthropogenic disturbance degree on the ecological risk of the landscape in the study area, with reference to previous research results [40,41] and combined with the actual situation in the study area, the following formula was calculated:

$$HD = \frac{\sum_{i=1}^{m} UI_i S_i}{S} \qquad (10)$$

where *HD* is the anthropogenic disturbance of the *i*th grid, $UI_i$ is the disturbance index of the ith landscape type, $S_i$ is the area of the *i*th landscape type (km$^2$), S is the total area of the grid cells (km$^2$), and m is the number of landscape types.

The factor detection and interaction detection functions in the geographic detector spatial analysis model [42,43] were selected to analyze the effects of individual factors on the study variables and the effects of multiple factor interactions on the variables. The six drivers' data collected were reclassified using ArcGIS 10.2 before being sampled separately, and the sampling results were imported into the geographic detector model for analysis, and were calculated as follows:

$$q = 1 - \frac{\sum_{h=1}^{L} N_k \sigma_k^2}{N \sigma^2} = 1 - \frac{SSW}{SST} \qquad (11)$$

where the value of *q* measures the impact of each driver on ERI and has a value range of (0–1). The larger the value, the greater the influence of the factor on ERI and the more pronounced the influence on the spatial analysis of ERI; h is the partition sequence the number of the independent variable; *L* is the total number of partitions; $N_k$ and *N* are the total number of rasters in each partition and the region, respectively; $\sigma_k^2$ and $\sigma^2$ are the variance of each partition and the variance of the ecological risk of the landscape in the region. *SSW* and *SST* are the sum of variance within layers and the total variance of the whole region, respectively.

## 3. Results

### 3.1. Spatial and Temporal Characteristics of LULC Change

The largest and most widely distributed LULC type in the study area was grassland, followed by cropland and bare land. Cropland, rivers, lakes, swamps, and construction land were mainly located on the northern bank of the Yellow River, while forests were concentrated in the northeastern part of the basin. Bare land was mainly in the middle and southeastern parts of the basin (Figure 2). From 1990 to 2020, the areas of grassland, bare land, and swamps showed an overall decreasing trend, decreasing by 1093.44 × 10$^4$ ha, 191.20 × 10$^4$ ha, and 16.72 × 10$^4$ ha, respectively. The areas of cropland, forest, and construction land increased by 17.09 × 10$^4$ ha, 29.46 × 10$^4$ ha, and 11.94 × 10$^4$ ha, respectively. The areas of rivers and lakes kept fluctuating but remained the same (Figure 3).

The LULC transfer from 1990 to 2020 was mainly reflected in the transformation of grassland and bare land into other land types (Table 1). Grassland and bare land lost 110.68 × 10$^4$ ha and 85.48 × 10$^4$ ha, respectively, and the area of both that transferred to cropland, forest land, rivers, lakes, and construction land was 89.76 × 10$^4$ ha, 63.15 × 10$^4$ ha, 2.70 × 10$^4$ ha, 1.70 × 10$^4$ ha, 2.98 × 10$^4$ ha, and 50.22 × 10$^4$ ha and 13.62 × 10$^4$ ha, 0.61 × 10$^4$ ha, 101.94 × 10$^4$ ha, 2.07 × 10$^4$ ha, 0.41 × 10$^4$ ha, 1.15 × 10$^4$ ha, and 10.28 × 10$^4$ ha.

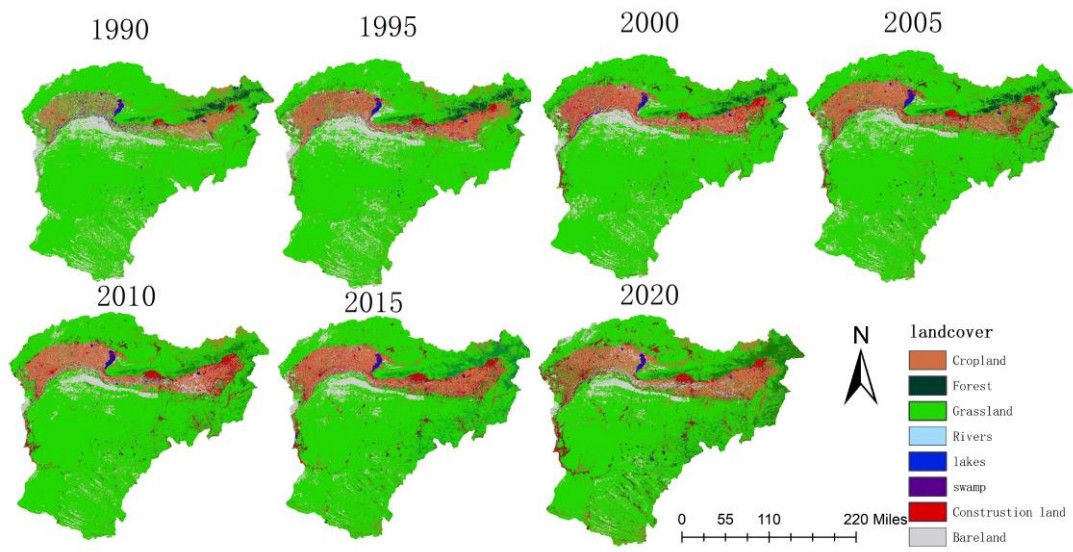

**Figure 2.** Distribution of main LULC types in the YRBIM from 1990 to 2020.

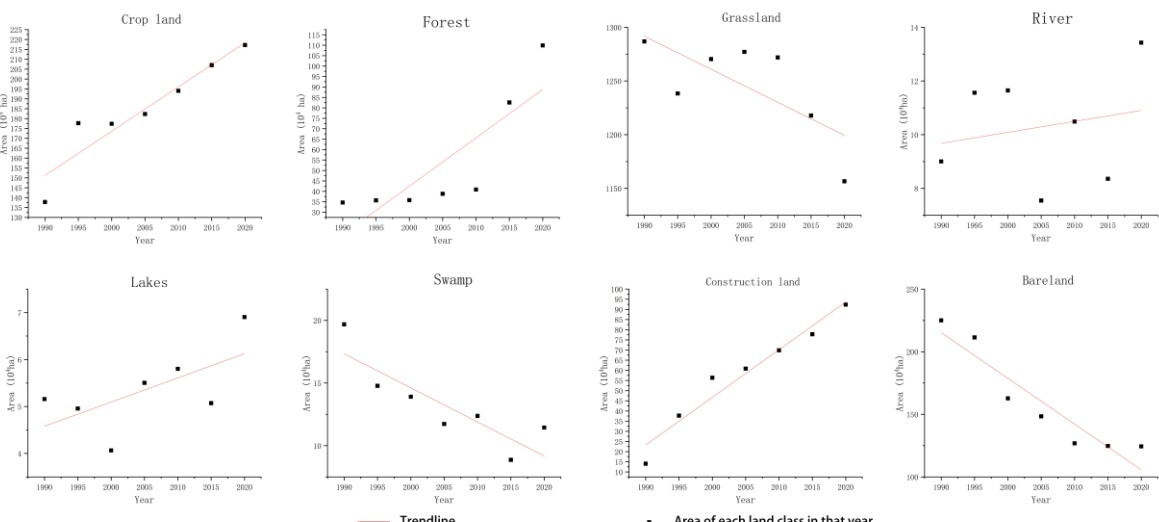

**Figure 3.** Changes in the area of each category from 1990 to 2020.

**Table 1.** LULC-type transition matrix from 1990 to 2020 ($10^4$ ha).

| 1990/2020 | Cropland | Forest | Grassland | Rivers | Lakes | Swamp | Construction Land | Bare Land | Total |
|---|---|---|---|---|---|---|---|---|---|
| Cropland | 72.16 | 4.77 | 26.49 | 1.00 | 0.16 | 1.32 | 7.66 | 3.53 | 117.09 |
| Forest | 1.51 | 24.46 | 2.59 | 0.15 | 0.04 | 0.19 | 0.51 | 0.01 | 29.46 |
| Grassland | 89.76 | 63.15 | 845.69 | 2.70 | 1.70 | 2.98 | 50.22 | 37.23 | 1093.44 |
| Rivers | 0.86 | 0.04 | 0.90 | 3.02 | 0.33 | 0.60 | 1.07 | 0.83 | 7.65 |
| Lakes | 0.14 | 0.04 | 0.24 | 0.36 | 2.64 | 0.51 | 0.39 | 0.06 | 4.37 |
| Swamp | 4.42 | 0.14 | 2.70 | 1.83 | 0.53 | 2.73 | 2.28 | 2.09 | 16.72 |
| Building | 2.05 | 0.18 | 2.22 | 0.28 | 0.06 | 0.24 | 6.05 | 0.85 | 11.94 |
| Bare land | 13.62 | 0.61 | 101.94 | 2.07 | 0.41 | 1.15 | 10.28 | 61.12 | 191.20 |
| Total | 184.53 | 93.38 | 982.76 | 11.41 | 5.87 | 9.72 | 78.48 | 105.72 | |

### 3.2. Spatial and Temporal Variation in Landscape Ecological Risk

The average ecological risk value between 1990 and 2020 for the overall YRBIM landscape were 0.024086, 0.02386, 0.023039, 0.022854, 0.022486, 0.022486, 0.022385, and 0.022516, respectively, with a decreasing trend in the ecological risk level. Lower risk areas

were the most widespread, whereas high and higher risk areas were mainly concentrated in the central and southeastern regions (Figure 4). The share of area in low risk areas continued to trend upwards, with an increase of $183,782.79 \times 10^4$ ha. The percentage of area in high and higher risk areas showed a decreasing trend, with a decrease of $80,857.45 \times 10^4$ ha and $99,008.01 \times 10^4$ ha, respectively, while the area share of low and medium risk areas remained the same, at approximately 10% and 17%, respectively (Table 2).

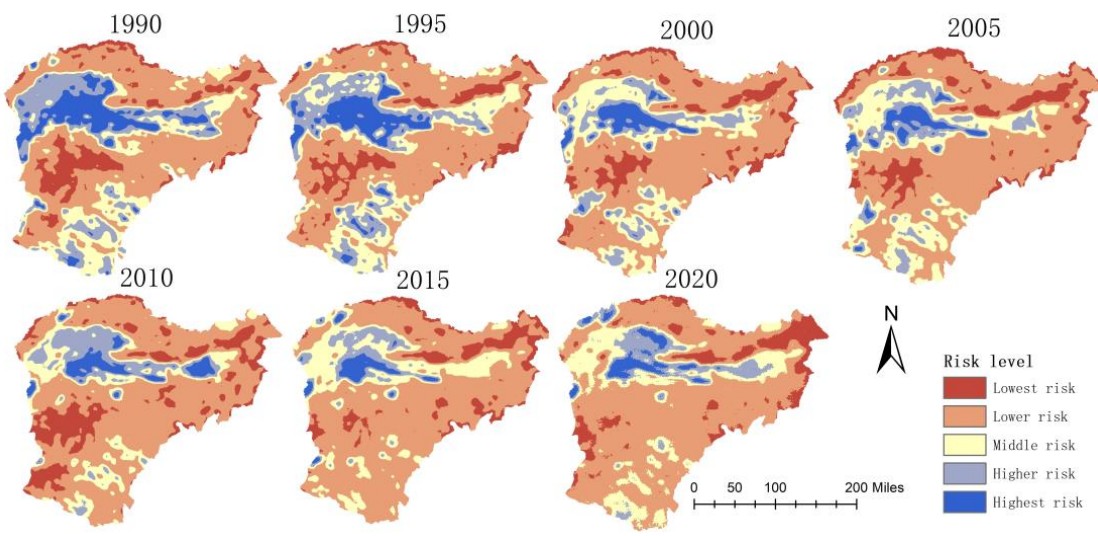

**Figure 4.** Landscape ecological risk classification from 1990 to 2020.

**Table 2.** Area and proportion of different risk levels.

| | Lowest Risk | | Lower Risk | | Middle Risk | | Higher Risk | | Highest Risk | |
|---|---|---|---|---|---|---|---|---|---|---|
| | Percent (%) | Area ($10^4$ ha) | Percent (%) | Area ($10^4$ ha) | Percent (%) | Area ($10^4$ ha) | Percent (%) | Area ($10^4$ ha) | Percent (%) | Area ($10^4$ ha) |
| 1990 | 10.86% | 143,869.68 | 46.59% | 617,184.18 | 17.65% | 233,826.30 | 15.35% | 203,295.78 | 9.55% | 126,554.94 |
| 1995 | 9.74% | 129,007.44 | 48.73% | 64,5529.50 | 18.73% | 248,102.10 | 14.38% | 190,527.75 | 8.42% | 111,564.09 |
| 2000 | 10.90% | 144,393.84 | 54.50% | 721,953.54 | 19.71% | 261,117.09 | 10.30% | 136,412.91 | 4.59% | 60,853.41 |
| 2005 | 11.75% | 155,696.58 | 54.59% | 723,161.34 | 20.03% | 265,330.89 | 9.76% | 129,247.83 | 3.87% | 51,294.25 |
| 2010 | 15.07% | 199,674.18 | 57.26% | 758,584.62 | 13.72% | 181,704.87 | 9.74% | 129,068.73 | 4.20% | 55,698.42 |
| 2015 | 9.81% | 129,933.09 | 62.66% | 830,081.25 | 18.01% | 238,548.51 | 6.51% | 862,24.19 | 3.01% | 39,917.73 |
| 2020 | 10.71% | 141,858.63 | 60.46% | 800,966.97 | 17.51% | 231,920.01 | 7.87% | 104,287.77 | 3.45% | 45,697.49 |

The global Moran's value for landscape ecological risk in the YRBIM from 1990 to 2020 ranged from to 0 to 1, with a more significant positive spatial correlation and aggregation effect. In addition, Moran's *I* index showed a decreasing trend from 1990 to 2020 despite fluctuations (Figure 5).

Hot spots in the YRBIM were mainly concentrated in the central and southeastern parts of the basin, whereas cold spots were mainly distributed in the northern, south-central, and western parts. Thirty years ago, the distribution of hot spot areas did not change considerably, whereas the cold spot areas decreased significantly in the northern and central parts of the basin after 2010 (Figure 6). The areas of both cold and hot spots showed a decreasing trend, decreasing by $184.39 \times 10^4$ ha and $39.58 \times 10^4$ ha, respectively (Figure 7).

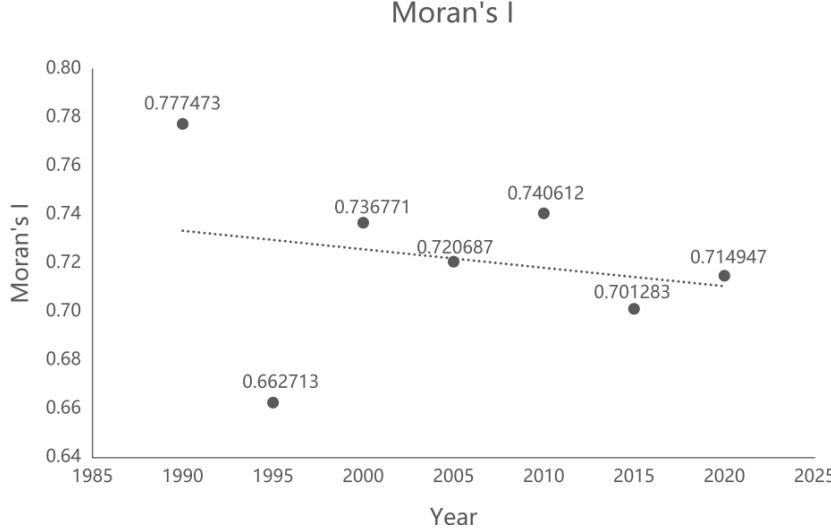

**Figure 5.** Moran's *I* index changes from 1990 to 2020.

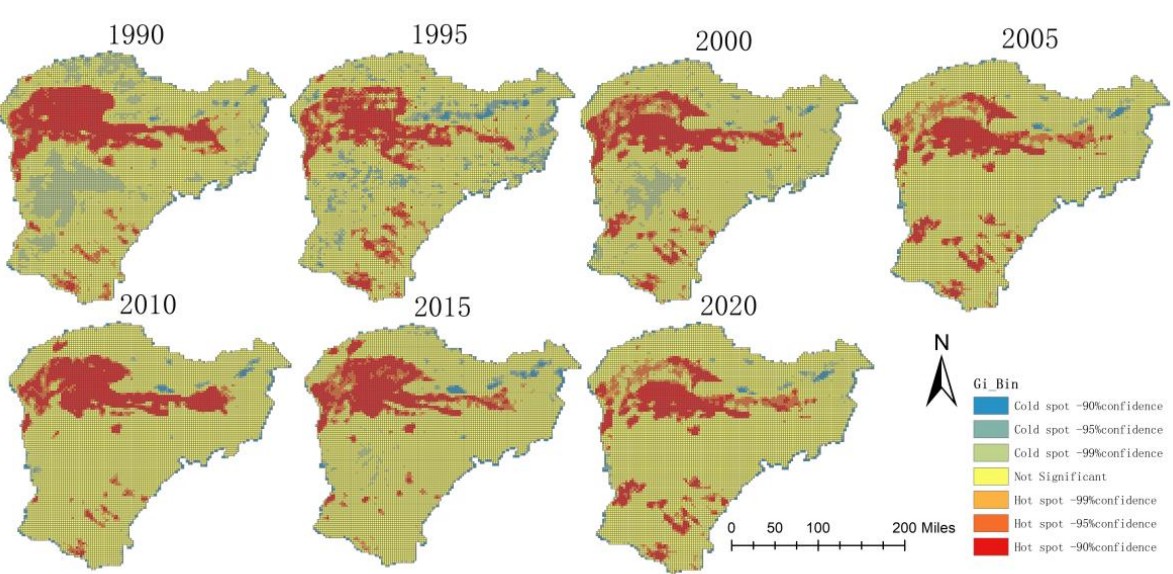

**Figure 6.** Analysis of hot and cold spots of landscape ecological risk from 1990 to 2020.

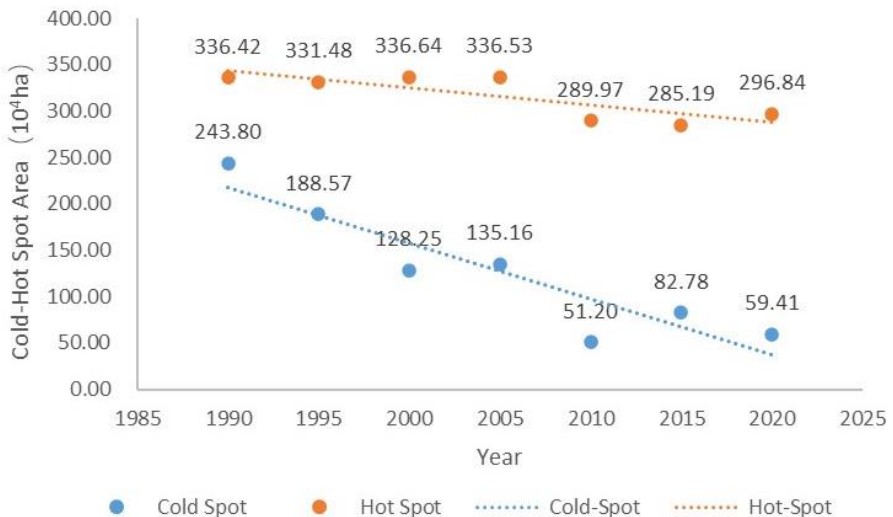

**Figure 7.** Changes in the area of landscape ecological risk cold and hot spots from 1990 to 2020 ($10^4$ ha).

### 3.3. Analysis of the Drivers of Change in Landscape Ecology Risk

Anthropogenic disturbance and elevation were the two main factors that strongly influenced changes in landscape ecological risk in the YRBIM (Figure 8). The contribution of any two of the factors listed is greater than the contribution of a solitary factor, with the interaction mainly being a two-factor enhancement or nonlinear enhancement. The interactions between anthropogenic and natural factors were stronger than those between natural factors (Figure 9).

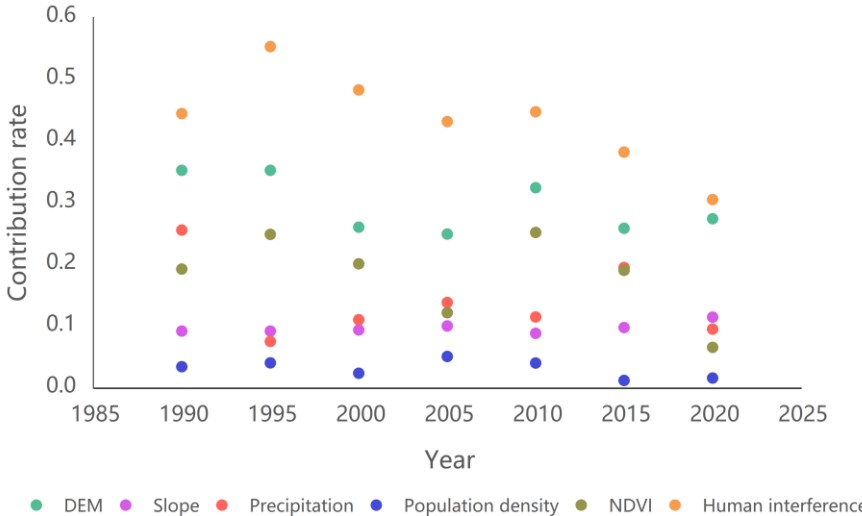

**Figure 8.** The explanatory power of landscape ecological risk drivers from 1990 to 2020.

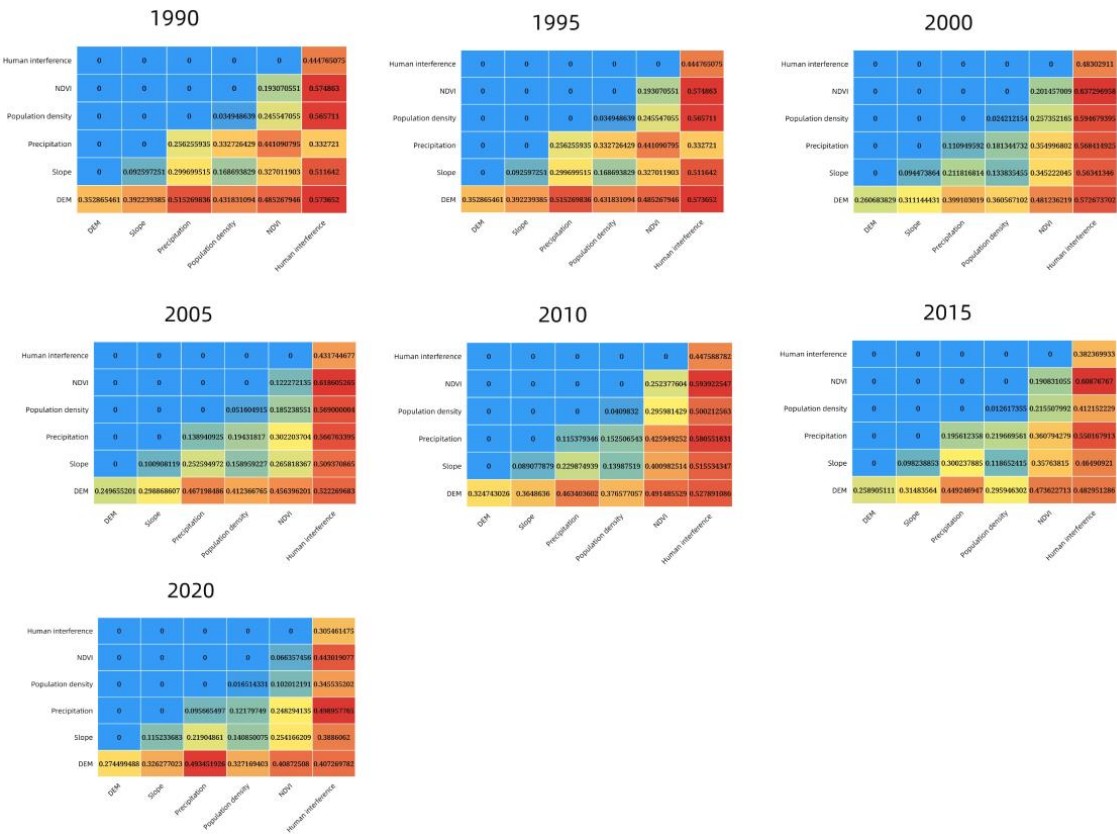

**Figure 9.** The interactive detection results of landscape ecological risk driving factors from 1990 to 2020.

## 4. Discussion

### 4.1. Characteristics of Land Type Transfer in the YRBIM

In the YRBIM, there is a concentration of land transfer from several land types to a few land types and from a few land types to other land types to varying degrees, showing a 'many-to-one' and 'one-to-many' pattern of transfer. This finding is consistent with the results of Qu [29] regarding LULC change in the entire Yellow River Basin. During the land transfer process, grasslands and bare land play an important role in "area transfer out". Grasslands and deserts are the most well-known landscape symbols in Inner Mongolia, and grassland and bare land are also two of the larger and more widespread land types in the basin, making them advantageous for transfer to other land types in the regional development process [44]. Cropland and construction land are important players in the process of land transfer, with "area transfer". The Hetao area in the northern part of the basin has assumed the burden of food production since ancient times and has an important strategic position [45] where food security has become a major concern [46]. In addition, The establishment and the development of the "Hohhot-Baotou-Ordos" economic circle in the basin [47] has made cropland and construction land the main targets for other land types to be transferred.

### 4.2. Ecological Risks in the YRBIM Show a Decreasing Trend

From 1990 to 2020, the ecological risk of the YRBIM decreased. The spatial distribution of ecological risk hot spot areas and cold spot areas concerning their ecological risk levels remained the same, and the hot spot areas also showed a decreasing trend. This is in agreement with the results of Du et al.'s [48] ecological risk evaluation of the Yellow River Basin in the last few years. The gradient distribution pattern of the risk levels was closely related to LULC changes [33]. From 1990 to 2020, the overall area of low risk areas in the study area increased and the area of high risk areas decreased, mainly owing to the large shift from grassland and bare land to cropland, construction land, and forest during the study period, and the landscape dominance increased. Landscape fragmentation and separation decreased significantly [18], leading to a shift in risk levels to lower levels. The ecological risk-sensitive areas (hot spot areas) of the basin mainly contain land types such as cropland, bare land, and construction land, which have a high degree of landscape fragility and strong human interference. Cold areas of ecological risk in the basin are mainly located in forests or grasslands where the level of human activity is comparatively low, with high vegetation cover and low landscape fragmentation, and where forests can improve the quality of soil and the stability of ecosystems. At the same time, government control and regulation of industrial pollutant emissions, pollution of water and soil resources, and rational planning of land type changes in recent decades have also led to an increase in ecosystem stability and a reduction in landscape fragmentation and separation.

### 4.3. Human Activity Is the Dominant Factor Driving Ecological Risk Change in the YRBIM

Topography, climate, and human activities jointly drive changes in ecological risk in the YRBIM [49], but human activities contribute to the dominant role of ecological risk change in this area, which is in agreement with the results of Deng et al. [50] in their exploration of the drivers of ecological risk in the Yellow River Basin. The greater the intensity of human activities, the more damage is done to the surface landscape, reducing its ecological stability. In contrast, the ecological risk was significantly lower at high altitudes than at low altitudes, as found by Hamed et al. [20] in their study on the drivers of ecological risk in the Dongjiang River Basin. The ecological integrity of the landscape was well-maintained because less landscape disturbance occurs at higher elevations. A range of ecological conservation and rehabilitation projects, such as the integrated soil and hydrological conservation project initiated in the 1990s, have led to increased land use [51], which reflects the effects of ecological civilization construction and ecological conservation [52] to a certain extent [42]. The people of Inner Mongolia have a long

history of living in harmony with nature, and the scientific concept of sustainable grassland exploitation dates back to the time of Genghis Khan [53].

*4.4. Policy Recommendations and Research Limitations*

The YRBIM plays a critical role in the general development and ecological security of the Yellow River Basin and the entire region [54]. With the progress of ecological civilization construction projects and the further strengthening of urbanization, the mechanism of coordinated regional development will improve [55]. However, issues such as landscape fragmentation and increased ecological risk due to unreasonable land development and utilization during the process of social and economic development pose considerable challenges. Therefore, it is crucial to follow the principle of protection against overproduction in the process of regional development: build a "mountain, water, field, forest, lake, grass, and sand" community of destiny [56], prioritize the impact of urban construction, agricultural development, and mineral resource development on the environment, strengthen land use planning in areas sensitive to ecological risks, and strictly control the scale of construction land. Future research in ecological risk management will focus on prevention before loss rather than recovery after loss, and we need to look at ways to enhancing the connectivity and integrity of regional landscapes, reducing the concentration of risky areas, and mitigating the pressure of ecological risks.

The LULC-based approach to landscape ecological risk assessment is a useful tool that can be applied to multiscale areas and can achieve spatial and temporal representation of multisource risks without extensive field observations [2,33]. However, there are several inconclusive factors in this study. First, the results of landscape ecological risk assessment are significantly reliant on the accuracy of the LULC. Therefore, inaccuracies in LULC data may cause uncertainty in the ecological risk assessment of the landscape [57]. Although the overall accuracy of the seven LULC datasets used in this study was higher than 80%, errors in LULC data cannot be eliminated. Therefore, increasing the accuracy of LULC data should be a priority for future research. Second, landscape ecological risk assessment is only considered for the landscape. In the future, the sustainable development of the YRBIM can be analyzed from multiple perspectives, including those related to human activities and economic factors [58]. Third, the landscape ecological risk assessment method only considers the proportion of the area of different landscape types and lacks the ecological meaning of ecological risk [33]. Despite these uncertainties, this study is significant in assessing the ecological risk in the YRBIM based on a more widely applied method that validates the effectiveness of conservation policies to some extent. This work can provide support for protecting the eco-quality of the basin.

## 5. Conclusions

This study analyzed the spatial and temporal changes in ecological risk and its drivers in the YRBIM based on LULC data from 1990 to 2020. We found that high ecological risk areas were primarily concentrated in the central part of the basin, in the Hetao Irrigation Basin and the Hobq Desert, as well as in the southeastern part of the basin, in the Mu Us Sandy Land. The ecological risk situation is gradually improving, with human activities being the main factor driving the changes in ecological risk in the basin. To further enhance environmental protection in the area, the Chinese government has enacted the Yellow River Protection Law and initiated several remediation projects aimed at enhancing the ecological sustainability of the Yellow River Basin. This study provides important reference material for the ecological restoration of the YRBIM, the construction of an ecological civilization, suggestions for global environmental governance, and the building of a community of life.

**Author Contributions:** Q.Z. conceptualized the problem; H.Z. and J.Z. specified the research methodology; H.Z., L.Y. and J.Z. analyzed the data using software; H.Z. validated the study; H.Z. conducted a formal analysis of the study; Q.Z. conducted the survey; H.Z. collated the data resources; H.Z. wrote and prepared the original manuscript; Q.Z. and N.Z. reviewed and edited the article; H.Z., Z.L. and J.Z. visualized the article data results; Q.Z. supervised; Q.Z. project managed; and Q.Z. secured funding. All authors have read and agreed to the published version of the manuscript.

**Funding:** This study was supported by the Major Program of Inner Mongolia (2020ZD0009), the National Key Research and Development Program of China (2021YFC3201201), the Cooperation Project of Science and Technology Promotion in Inner Mongolia (2022EEDSKJXM002-1), and the Grassland Talents Program of Inner Mongolia (CYYC9013).

**Data Availability Statement:** Not applicable.

**Acknowledgments:** We are grateful to Zhang Qing of Inner Mongolia University, China, for his generous support of this research.

**Conflicts of Interest:** There is no conflict of interest in this study.

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
