# Peer review of "Spatio-Temporal Assessment of Landscape Ecological Risk and Associated Drivers: A Case Study of the Yellow River Basin in Inner Mongolia"

_land, doi:10.3390/land12061114_

Round 1

Reviewer 1 Report

This study attempts to analyze the ecological risk assessment in the Inner Mongolia region of the Yellow River Basin from 1990 to 2020. This topic is practical, however, the innovation and contribution of this study should be further strengthened. The chosen of the case study in Inner Mongolia is suggested to be mentioned, and how to extend the results to some other provinces is essential. Some comments and suggestions are listed as below for reference.

The literature review should be further enhanced, especially through comparing with previous studies focusing on ecological risks.

How to define and calculate the Landscape Ecological Risk should be clearly mentioned and discussed.

The method to analyze different drivers of landscape ecology risk in Section 3.3 should be listed in the Method section.

The reason to choose the United States Geological Survey (USGS) instead of those generated by Chinses organizations/scholars should be mentioned, including the potential disparity and remaining uncertainty.

A separate table listing the ecological risk assessments of different regions and typical areas should be compared and discussed in Section 4.

The discussion and policy implications should be enhanced, and how to explain the decreasing trend in the ecological risk level during 1990 and 2020 should be clarified.

Some practical policy implications are suggested based on the results derived in this study, and buffer zones should be reserved to protect the ecological environmentmentioned in the discussion is not supportive.

The uncertainty of the results should be discussed based on some qualified data.

The quality of figures such as Figs 3 and 5 should be improved, and the changes in Figs 2 and 6 are not obvious.

The format of references needs to be systematically checked, and the decimal points of the results in the manuscript and Tables needs to be unified.

The language of the manuscript needs to be polished throughout.

Moderate editing of English language is suggested.

Author Response

Thank you for your constructive comments, which we feel are invaluable to our research and to the writing of our scientific papers. I have submitted the specific changes to the system for your review. Many Thanks again!

Reviewer 2 Report

Dear authors.

I am grateful for the opportunity to review the article "Spatio-temporal Assessment of Landscape Ecological Risk and Associated Drivers: A Case Study of the Yellow River Basin in Inner Mongolia". The article is devoted to a topical research topic that has been studied by many authors in recent years in various regions of the world. The article has a lot of merit. It is worth noting the calculations performed by the authors and the illustrative material presented.

However, the article has a few remarks.

1 Figure 1, 3 is not readable. Improve image quality.

2 Check line design 396 410

3 In section 1, it is worth adding the history of the study of the issue in other countries of the world

4 Specify study limitations

The article can be accepted only after the comments have been eliminated.

Author Response

(The authors gave the same response as above.)

Reviewer 3 Report

The paper titled Spatio-temporal Assessment of Landscape Ecological Risk and Associated Drivers: A Case Study of the Yellow River Basin in Inner Mongolia represents a major contribution to the field of ecosystem risk assessment, land use, and land cover changes. In this study, the authors constructed a landscape ecological risk index (ERI) model based on changes in LULC to assess the ecological risk of the basin and analyze the driving factors of its spatial heterogeneity using a geographic detector model. This study provides an important reference for ecological restoration and civilization construction in the Yellow River Basin in Inner Mongolia, which is very important for further research.

This topic is very important, original, and relevant in this field of research.

The methodology used is adequate. Also, the quotations are relevant, and the references are appropriate. The research design is appropriate, and the methods are adequately described. The results are presented adequately and comprehensibly. All figures and tables are well-presented and clear.

The conclusion is supported by the results, and they are consistent with the evidence and arguments presented in the manuscript.

Some technical issues:

Figure 3 (Changes in the area of each category from 1990 to 2020) and Figure 9 (The interactive detection results of landscape ecological risk driving factors from 1990 to 231 2020) are less visible and it is necessary to intensify the color.

Put one line (space) before line 204.

Pay attention to the page break so that the names of the figures can be seen immediately below and above. The same goes for headlines. The title is at the bottom of one page and the text is on the next.

The article is acceptable for publication in Land after minor revision.

Author Response

Thank you for your constructive comments, which we feel are very important to us. We have uploaded the comment response letter to our system. Thank you again! Please see the attachment.
